StrainFLAIR: strain-level profiling of metagenomic samples using variation graphs

Da Silva Kévin 1 2 kevin.da-silva@inria.fr
http://orcid.org/0000-0003-1926-8077 Pons Nicolas 1
http://orcid.org/0000-0002-6762-5350 Berland Magali 1
Plaza Oñate Florian 1
http://orcid.org/0000-0003-4971-0049 Almeida Mathieu 1
http://orcid.org/0000-0003-0776-6407 Peterlongo Pierre 2
1 Université Paris-Saclay, INRAE, MGP , Jouy-en-Josas , France
2 Univ Rennes, Inria, CNRS, IRISA—UMR 6074 , Rennes , France
Gillespie Joseph
Electronic publication date: 2021 Aug 23
Publication date: 2021
Volume: 9
Electronic Location ID: e11884
Received 2021 Feb 15; Accepted 2021 Jul 9
Copyright: © 2021 Da Silva et al.
Copyright year: 2021
Copyright holder: Da Silva et al.
License: This is an open access article distributed under the terms of the Creative Commons Attribution License, which permits unrestricted use, distribution, reproduction and adaptation in any medium and for any purpose provided that it is properly attributed. For attribution, the original author(s), title, publication source (PeerJ) and either DOI or URL of the article must be cited.
License URL: https://creativecommons.org/licenses/by/4.0/

Keywords: Metagenomics, Variation graphs, Strain-level abundances, Read mapping

Funding: French INRAE HoloFlux Brittany Region, France This work was supported by a grant of the French INRAE HoloFlux metaprogram and by the Brittany Region, France. The funders had no role in study design, data collection and analysis, decision to publish, or preparation of the manuscript.

==============================
Current studies are shifting from the use of single linear references to representation of multiple genomes organised in pangenome graphs or variation graphs. Meanwhile, in metagenomic samples, resolving strain-level abundances is a major step in microbiome studies, as associations between strain variants and phenotype are of great interest for diagnostic and therapeutic purposes. We developed StrainFLAIR with the aim of showing the feasibility of using variation graphs for indexing highly similar genomic sequences up to the strain level, and for characterizing a set of unknown sequenced genomes by querying this graph. On simulated data composed of mixtures of strains from the same bacterial species Escherichia coli, results show that StrainFLAIR was able to distinguish and estimate the abundances of close strains, as well as to highlight the presence of a new strain close to a referenced one and to estimate its abundance. On a real dataset composed of a mix of several bacterial species and several strains for the same species, results show that in a more complex configuration StrainFLAIR correctly estimates the abundance of each strain. Hence, results demonstrated how graph representation of multiple close genomes can be used as a reference to characterize a sample at the strain level.

Introduction

The use of reference genomes has shaped the way genomics studies are currently conducted. Reference genomes are particularly useful for reference guided genomic assembly, variant calling or mapping sequencing reads. For the latter, they provide a unique coordinate system to locate variants, allowing to work on the same reference and easily share information. However, the usage of reference genomes represented as flat sequences reaches some limits (Ballouz, Dobin & Gillis, 2019). One sequence chosen as the reference among other homologous sequences does not capture the whole genomic variability. Hence, reads from non-reference alleles may be mis-mapped or not mapped at all. Secondly, with the increasing availability of new genomes, several sequences can be used as multiple references. However, close genomes (typically genomes of strains of the same species) show a high sequence similarity. The mapping of sequencing reads results in mis-mapped reads or ambiguous alignments generating noise in the downstream analysis (Na et al., 2016).

This has led recent methods to provide a representation of multiple genomes as genome graphs, also called variation graphs, in which each path is a different known variation. Such graph representations are well defined, and tools to build and manipulate graphs are under active development (Garrison et al., 2017; Kim et al., 2019; Rakocevic et al., 2019; Li, Feng & Chu, 2020). This graph structure provides obvious advantages such as the reduction of the data redundancy, while highlighting variations (Garrison et al., 2018). However, it also introduces novel difficulties. Updating a graph with novel sequences, adapting existing efficient algorithms for read mapping, and, mainly, developing new ways to analyse sequence-to-graph mapping results for downstream analyses are among those new challenges. The work presented here primarily focuses on this latest point. It proposes to show the feasibility of using variation graphs for profiling metagenomic samples at the strain level, that is to say identifying and estimating abundances of strains contained in a metagenomic sample.

In the context of metagenomics, representing genomes in graphs is of particular interest for indexing microorganism genomes. Microorganisms are predominant in almost every ecosystems from ocean water (Sunagawa et al., 2015) to human body (Clemente et al., 2012), and play major functioning roles in them (New & Brito, 2020). While studies in microbial ecology are facing a bottleneck due to the difficulty of isolating and cultivating most of those microbes in laboratory, preventing the analysis of the complex structure and dynamics of the microbial communities (Stewart, 2012), high-throughput sequencing in metagenomics offers the opportunity to study a whole ecosystem. In particular, shotgun sequencing allows a resolution up to the species level (Jovel et al., 2016), and enables samples analysis in terms of population stratification, microbial diversity or bio-markers identification (Quince et al., 2017b). Understanding of microbial communities structure and dynamics is usually revealed by resolving the species present in samples and their relative abundances, which can then be associated with phenotypes, notably in the field of human health (Ehrlich, 2011, Vieira-Silva et al., 2020; Solé et al., 2021). Characterizing samples at the strain level has a growing interest, as it may highlight new associations with phenotypes. A better understanding of the functional impact of strains in host-microbe interactions is crucial to new therapeutic strategies and personalized medicine. Escherichia coli, which has a highly variable genome, is a well-known example since some strains are harmless commensals in the human gut microbiota while others are harmful pathogens (Rasko et al., 2008; Loman et al., 2013). Current approaches using gene catalog handle multiple similar genomes by selecting a representative sequence from cluster of genes, thus getting rid of the redundancy but also of the variations, yet crucial to distinguish the strains of a species (Qin et al., 2010).

Although they are not based on a graph representation of the reference genomes, several tools have already been developed this last few years to study the strain composition of metagenomic samples. DESMAN (Quince et al., 2017a) and mixtureS (Li, Hu & Li, 2020) use known core genes from the species of interest and a single reference genome, respectively. Using those data as references, and from sequencing reads, these methods infer non-identified haplotypes, defining them as de novo approaches. Additionally, DESMAN operates on a multiple set of sequencing reads. PanPhlan (Scholz et al., 2016) which uses a set of reference genomes and StrainPhlan (Truong et al., 2017) which uses markers from reference genomes are complementary tools providing a gene family presence/absence matrix and strain identification only for the dominant strain, respectively. StrainEst (Albanese & Donati, 2017) and DiTASiC (Fischer, Strauch & Renard, 2017) use a set of reference genomes, providing abundance estimation of strains present in the sample. Finally, while designed for metagenomics classification, Kraken2 (Wood, Lu & Langmead, 2019) and KrakenUniq (Breitwieser, Baker & Salzberg, 2018), which can use a custom database of reference genomes, offer meaningful outputs to characterize metagenomic samples. Those tools are further discussed in this article alongside the result they provide.

In this work, we present StrainFLAIR, a novel method and its implementation that uses variation graph representation of gene sequences for strain identification and quantification. We proposed novel algorithmic and statistical solutions for managing ambiguous alignments and computing an adequate abundance metric at the graph node level. Results on simulated data and on real sequencing data have shown that we could correctly identify and quantify strains present in a sample. Notably, in the controlled experimental design that we investigated, we could also detect the existence of a strain close to, but absent from those in the reference.

StrainFLAIR is available at http://github.com/kevsilva/StrainFLAIR.

Methods

We propose here a description of our tool StrainFLAIR (STRAIN-level proFiLing using vArIation gRaph). This method exploits various state-of-the-art tools and proposes novel algorithmic solutions for indexing bacterial genomes at the strain level. It also permits to query metagenomes for assessing and quantifying their content, in regards to the indexed genomes. An overview of the index and query pipelines are presented on Fig. 1.

Figure 1 StrainFLAIR overview.

(A) Indexing. Input is a set of known reference genomes of various bacterial species and strains. StrainFLAIR uses a graph for indexing genes of those reference genomes. (B) Read mapping on the previously mentioned graph. (C) Mapped reads analysis. StrainFLAIR assigns and estimates species and strain abundances of a bacterial metagenomic sample represented as short reads.

Rational for the choice of third-party tools and their detailed usages are given in Section S1.1.

In a few words, StrainFLAIR works as follows: First, it indexes genes of input reference genomes. Similar genes from several genomes are grouped into a gene family. Each gene family is represented as a part (a connected component) of a variation graph. The path described in this variation graph by the sequence of any gene of any indexed genome is called a “colored-path”. Note that, conversely, any path of the variation graph does not necessarily correspond to an indexed gene. At query time, the mapping of a queried read on the graph results on a subset of the graph in which each mapped nodes is associated with a mapping score. This set of nodes is called a “multipath-alignment”. From a multipath-alignment we extract a set of so called “single-path-alignments” that are paths with a mapping score higher than a threshold. Then, in a step called “colored-path attribution”, each of the previously determined single-path-alignments is, when possible, attributed to the most probable colored-path of the variation graph, hence determining to which input genome the mapped read belongs to. Once all read are mapped, the careful analysis of mapped colored-paths enables to draw a profile to the queried metagenomic sample.

We now provide more details on each of the StrainFLAIR steps.

Indexing strains

Gene prediction

As non-coding DNA represents 15% in average of bacterial genomes and is not well characterized in terms of structure, StrainFLAIR focuses on protein-coding genes in order to characterize strains by their gene content and nucleotidic variations of them. Moreover, non-coding DNA regions can be highly variable (Thorpe et al., 2017) and taking into account complete genomes would then lead to highly complex graphs, and combinatorial explosions when mapping reads. Additionally, complete genomes are not always available. Focusing on the genes allows to use also drafts and metagenome-assembled genomes or a pre-existing set of known genes (Qin et al., 2010; Li et al., 2014). Hence, StrainFLAIR indexes genes instead of complete genomes in graphs.

Genes are predicted using Prodigal, a tool for prokaryotic protein-coding genes prediction (Hyatt et al., 2010).

Knowing that some reads map at the junction between the gene and intergenic regions, by conserving only gene sequences, mapping results are biased towards deletions and drastically lower the mapping score. In order to alleviate this situation, we extend the predicted gene sequences at both ends. Hence, StrainFLAIR conserves predicted genes plus their surrounding sequences. By default, and if the sequence is long enough, we conserve 75 bp on the left and on the right of each gene.

Gene clustering

Genes are clustered into gene families using CD–HIT (Li & Godzik, 2006). For the clustering step, the genes without extensions are used in order to strictly cluster according to the exact gene sequences and no parts of intergenic regions. CD–HIT–EST is used to realize the clustering with an identity threshold of 0.95 and a coverage of 0.90 on the shorter sequence. The local sequence identity is calculated as the number of identical bases in alignment divided by the length of the alignment. Sequences are assigned to the best fitting cluster verifying these requirements.

Graph construction

Each gene family is represented as a variation graph (Fig. 2). Variation graphs are bidirected DNA sequence graphs that represents multiple sequences, including their genetic variation. Each node of the graph contains sub-sequences of the input sequences, and successive nodes draw paths on the graph. Paths corresponding to reference sequences are specifically called “colored-paths”. Each colored-path corresponds to the original sequences of a gene in the cluster.

Figure 2 Illustration of a variation graph structure and colored-paths.

Each node of the graph contains a sub-sequence of the input sequences and is integer-indexed. A path corresponding to an input sequence is called a colored-path, and is encoded by its succession of node ids, e.g. 1, 3, 5, 6 for the colored-path 1 in this example.

In the case of a cluster composed of only one sequence, vg toolkit (Garrison et al., 2017) is used to convert the sequence into a flat graph. Alternatively, when a cluster is composed of two sequences or more, minimap2 (Li, 2018) is used to generate pairwise sequence alignments. Then seqwish (Garrison, 2021) is used to convert these pairwise sequence alignments into a variation graph. All the so-computed graphs (one per input cluster) are then concatenated to produce a single variation graph where each cluster of genes is a connected component.

The index is created once for a set of reference genomes. Afterward, any set of sequenced reads can be queried at the strain-level based on this index.

Querying variation graphs

The so-created variation graphs is queried by reads. Each read is mapped on the graph. Then each mapped read is associated, when possible, to a gene of one of the indexed genome. This is the “read attribution” step, itself composed of the “single-path-alignments attribution” and the “colored-path attribution” steps, detailed below.

Mapping reads

For mapping reads on the previously described reference graph, we used the sequence-to-graph mapper vg mpmap from vg toolkit. It produces a so-called “multipath-alignment”. A multipath-alignment is a graph of partial alignments and can be seen as a sub-graph (a subset of edges and vertices) of the whole variation graph (see Fig. 3 for an example). The mapping result describes, for each read, the nodes of the variation graph traversed by the alignment and the potential mismatches or indels between the read and the sequence of each traversed node.

Figure 3 Illustration of the multipath-alignment concept and the read attribution process.

The region of the read in blue aligns un-ambiguously to a node of the graph while the dark and light red parts can either align to the top or the bottom nodes of their respective mapping localization (due to mismatches that can align on both nodes for example), drawing an alignment as a sub-graph of the reference variation graph, and thus opening the possibility of four single-path-alignments. (A) Single-path-alignments attribution. First, from the multipath-alignment (all four read sub-paths), the breadth search finds the possible corresponding single-path-alignment(s) while respecting the mapping score threshold imposed by the user. Here, for the example, all four possible paths are considered valid. (B) Colored-path attribution. Second, each single-path-alignment is compared to the colored-paths from the reference variation graph. Two single-path-alignments matched the colored-paths (4–6–8 and 5–6–7). As it mapped equally more than one colored-path, this read is not processed during the first step of the algorithm which focuses on reads mapping uniquely on a single colored-path, but falls in the multiple mapped reads case which is processed during the second step and will be considered shared by both matched colored-paths.

Reads attribution

When mapping a read on a graph with colored-paths, two key issues arise, as illustrated on Fig. 3. As mapping generates a sub-graph per mapped read, the most probable mapped path(s) have to be defined. Meanwhile, the most probable mapped path(s) corresponding to a colored-path also have to be defined. Hence we developed an algorithm to analyse and convert, when possible, a mapping result into one or several single-path-alignment(s) (successive nodes joined by only one edge) per mapped read. In addition we propose an algorithm to attribute each such single-path-alignment to most probable colored-path(s).

Single-path-alignments attribution

A breadth first search on the multipath-alignment is proposed. It starts at each node of the alignment with a user-defined threshold on the mapping score. A single-path-alignment with a mapping score below this threshold is ignored, and the single-path-alignment with the best mapping score is retained. Additionally, for each alignment, nodes are associated with a so-called “horizontal coverage” value. The horizontal coverage of a node by a read corresponds to the proportion of bases of the node covered by the read. Hence, a node has an horizontal coverage of 1 if all its nucleotides are covered by the read with or without mismatches or indels.

Because of possible ties in mapping score, the search can result in multiple single-path-alignments, as illustrated Fig. 3A. This situation corresponds to a read which sequence is found in several different genes or to a read mapping onto the similar region of different versions of a gene.

To take into account ambiguous mapping affectations, as shown below, the parsing of the mapping output is decomposed into two steps. The first step processes the reads that mapped only a unique colored-path (called “unique mapped reads” here), corresponding to a single gene. The second step processes the reads with multiple alignments (called “multiple mapped reads” here).

Colored-path attribution

Once a read is assigned to one or several single-path-alignment(s), it still has to be attributed, if possible, to a colored-path. The following process attributes each mapped read to a colored-path and various metrics for downstream analyses are computed. In particular, an absolute abundance for each node of the variation graph, called the “node abundance”, is computed, first focusing on unique mapped reads (first step). For a given single-path-alignment, the successive nodes composing this path are compared to the existing colored-paths of the variation graph. If the alignment matches part of a colored-path, the number of mapped reads on this path is incremented by one (i.e. reads raw count). The node abundance for each node of the alignment is incremented with its horizontal node coverage defined by this alignment. Alignments with no matching colored-paths are skipped.

Then, we focus on multiple mapped reads (second step), as illustrated Fig. 3B. During this step, a single-path-alignment matches multiple colored-paths. Hence, the abundance is distributed to each matching colored-path relatively to the ratio between them. This ratio is determined from the reads raw count of each path from the first step. For example, if 70 unique mapped reads were found for path1 and 30 for path2 during the first step, a read matching ambiguously both path1 and path2 during the second step counts as 0.7 for path1 and 0.3 for path2. This ratio is applied to increment both the raw count of reads and the coverage of the nodes.

Gene-level and strain-level abundances

StrainFLAIR output is decomposed into an intermediate result describing the queried sample and gene-level abundances, and the final result describing the strain-level abundances.

Gene-level

After parsing the mapping result, the first output provides information for each colored-path, i.e. each version of a gene. Thereby, this first result proposes gene-level information including abundances. Exhaustive description of these intermediate results is provided in Section S1.2. We describe here three major metrics outputted by StrainFLAIR:

The mean abundance of the nodes composing the path. Instead of solely counting reads, we make full use of the graph structure and we propose abundances computation for each node as previously explained, and as already done for haplotype resolution (Baaijens et al., 2019). Hence, for each colored-path, the gene abundance is estimated by the mean of the nodes abundance.

In order to not underestimate the abundance in case of a lack of sequencing depth (which could result in certain nodes not to be traversed by sequencing reads), the mean abundance without the nodes of the path never covered by a read is also outputted.

The mean abundance with and without these non-covered nodes are computed using unique mapped reads only or all mapped reads.

The ratio of covered nodes, defined as the proportion of nodes from the path which abundance is strictly greater than zero.

Strain-level

A colored-path associated to only one strain is called “strain-specific”. Strain-level abundances are obtained by exploiting the results of reads mapped on strain-specific colored-paths.

First, for each genome, the proportion of detected genes is computed, as the proportion of specific genes on which at least one read maps. Then, the global abundance of the genome is computed as the mean or median of all its specific gene abundances. However, if the proportion of detected genes is less than a user-defined threshold, the genome is considered absent and hence its abundance is set to zero.

StrainFLAIR final output is a table where each line corresponds to one of the reference genomes, containing in columns the proportion of detected specific genes, and our proposed metrics to estimate their abundances (using mean or median, with or without never covered nodes as described for the gene-level result).

Results presented Section S1.3 validate and motivate the proposed abundance metric by comparing it to the expected abundances and other estimations using linear models.

Results

We validated our method on both a simulated and a real dataset. All computations were performed using StrainFLAIR, version 0.0.1, with default parameters. The relative abundances estimation was based on the mean of the specific gene abundances, computed by taking into account all the nodes (including non-covered nodes), and using a 50% threshold on the proportion of detected specific genes.

The presented results are compared to Kraken2 considered as one of the state-of-the-art tool dedicated to the characterization of read set content, and based on flat sequences as references. Read counts given by Kraken2 were normalized by the genome length and converted into relative abundances. Other tested tools either suffer from unfair comparisons as their features differ from StrainFLAIR (DESMAN, PanPhlan and StrainPhlan) or show weaker results than those obtained by Kraken2 (StrainEst, DiTASiC, KrakenUniq and mixtureS). All results obtained with these tools are presented in Section S1.8.

Here we present a proof of concept of the variation graph application for the microbial strain detection. While the aim of this article is not to provide a benchmark of the state-of-the-art tools, computing setup and performances are indicated in Section S1.4.

Validation on a simulated dataset

We first validated our method on simulated data, focusing on a single species with multiple strains. Our aim was to validate the StrainFLAIR ability to identify and quantify strains given sequencing data from a mixture of several strains of uneven abundances, and with one of them absent from the index. Results presented in this section can be reproduced using data and commands available from the github website.

Reference variation graph

We selected complete genomes of Escherichia coli, a predominant aerobic bacterium in the gut microbiota (Tenaillon et al., 2010), and a species known for its phenotypic diversity (pathogenicity, antibiotics resistance) mostly resulting from its high genomic variability (Dobrindt, 2005).

Eight strains of E. coli were selected for this experiment from the NCBI (https://www.ncbi.nlm.nih.gov/genome/?term=txid562[orgn]). Seven were used to construct a variation graph (E. coli IAI39, O104:H4 str. 2011C-3493, str. K-12 substr. MG1655, SE15, O157:H16 str. Santai, O157:H7 str. Sakai, O26 str. RM8426), and one was used as an unknown strain in a strains mixture (E. coli BL21-DE3). For ease of reading, in the following, K-12 substr. MG1655 is simply designed by “K12” and BL21-DE3 is designed by “BL21”.

Mixtures and sequencing simulations

Our aim was to simulate the co-presence of several E. coli strains. Mixtures of three strains were used to mimic complex single species composition in metagenomic samples. We simulated short sequencing reads of 150 bp using vg sim from vg toolkit with a probability of sequencing errors set to 0.1%. Two batches of simulations were conducted in order to highlight the detection and quantification of strains in the mixture. The first simulation was a mixture composed of strains indexed in the reference graph (O104:H4, IAI39 and K-12) while the second simulation (O104:H4, IAI39 and BL21) had one absent from the reference variation graph (BL21) thus simulating a strain absent from the reference graph to be identified and quantified. For each simulation, we tested our StrainFLAIR with various read coverage (Table 1), with K-12 or BL21 in equal abundance of IAI39, potentially making it more difficult to distinguish, or in lower abundance, potentially making it more difficult to detect at all.

Table 1 Composition of the mixtures described in number of reads simulated and the corresponding coverage (in parentheses).

For each simulation (including either K-12, indexed in the variation graph, or BL21, not indexed), seven mixtures were simulated.

Samples	O104:H4	IAI39	K-12 or BL21	
1			200,000 (6.5×)	
2			100,000 (3×)	
3			50,000 (1.6×)	
4	300,000 (8.5×)	200,000 (5.8×)	25,000 (0.8×)	
5			10,000 (0.3×)	
6			5,000 (0.2×)	
7			1,000 (0.03×)	

Strain-level abundances

As explained in Methods, we computed the strain-level abundances using the specific gene-level abundance table obtained by mapping the simulated reads onto the variation graph. We compared our results to the expected simulated relative abundances.

Simulation 1: mixtures with K-12, present in the reference graph

StrainFLAIR successfully estimated the relative abundances of the three strains present in the mixture (Table 2), the sum of squared errors between the estimation given by our tool and the expected relative abundance was between 25 and 45 for all the experiments. However, it did not detect the very low abundant strain in the case of the mixture with 1,000 simulated reads for K-12 (coverage of ≈ 0.03×). With our methodology, the threshold on the proportion of detected genes (see Methods) lead to set relative abundance to zero of likely absent strains. This reduces both the underestimation of the relative abundances of the present strains and the overestimation of the absent strains.

Table 2 Reference strains relative abundances expected and computed by StrainFLAIR or Kraken2 for each simulated experiment with variable coverage of the K-12 strain.

Best results are shown in bold. For StrainFLAIR, the proportion of specific genes detected is shown in parentheses. Complete results are presented Section S1.6.

#reads
K-12	Method	O104:H4	IAI39	K-12	Sakai	SE15	Santai	RM8426	
1,000	Expected	59.88	39.92	0.2	0	0	0	0	
StrainFLAIR	56.47	43.53	0	0	0	0	0	
(0.995)	(0.989)	(0.309)	(0.189)	(0.151)	(0.188)	(0.212)	
Kraken2	38.91	60.72	0.22	0.04	0.07	0.03	0.02	
25,000	Expected	57.14	38.1	4.76	0	0	0	0	
StrainFLAIR	52.14	40.58	7.27	0	0	0	0	
(0.994)	(0.989)	(0.878)	(0.208)	(0.153)	(0.215)	(0.234)	
Kraken2	37.23	58.1	4.51	0.04	0.07	0.03	0.02	
200,000	Expected	42.86	28.57	28.57	0	0	0	0	
StrainFLAIR	38.12	29.81	32.08	0	0	0	0	
(0.993)	(0.988)	(0.99)	(0.211)	(0.159)	(0.219)	(0.237)	
Kraken2	28.31	44.18	27.35	0.04	0.08	0.03	0.02	

In comparison, Kraken2 did not provide this resolution. Applied to our simulated mixtures, while Kraken2 was slightly better for K-12 abundance estimation, it overestimated IAI39 relative abundance and underestimated O104’s one, leading to an overall higher sum of squared errors (between 456 and 872) compared to the expected abundances. Moreover, it set relative abundances to all the seven reference strains whereas four of them were absent from the mixture. This was expected as some reads (from intergenic regions for example) can randomly be similar to regions of genes from absent strains.

Simulation 2: mixtures with BL21, absent from the reference graph

Here, BL21 was considered an unknown strain, not contributing to the variation graph. The closest strain of BL21 in the graph, according to fastANI (Jain et al., 2018), was K-12 (98.9% of identity, see Section S1.5). Thus we expected to find signal of BL21 through the results on K-12.

As with the K-12 mixtures, StrainFLAIR successfully estimated the relative abundances of the two known strains present in the mixture (Table 3), the sum of squared errors between the estimation given by our tool and the expected relative abundance was between 22 and 180 for all the experiments. Labelled as K-12, it also gave close estimations for BL21 in this controlled experimental design. Again, it did not detect the very low abundant strain in the case of the mixture with 1,000, 5,000, and 10,000 simulated reads for BL21. Also similarly to the K-12 mixtures experiments, Kraken2 overestimated IAI39 relative abundance and underestimated O104’s one (sum of squared errors between 751 and 873), even less precisely than in the previous experiment. With sufficient coverage (here from the 0.8x for BL21), StrainFLAIR was closer to the expected values for all the reference strains than Kraken2.

Table 3 Reference strain relative abundances expected and computed by StrainFLAIR or Kraken2 for each simulated experiment with variable coverage of the BL21 strain, absent from the reference variation graph.

BL21 strain expected abundances are followed by an asterisk in the K-12 column. Best results are shown in bold. For StrainFLAIR, the proportion of specific genes detected is shown in parentheses. Complete results are presented Section S1.6.

#reads
BL21-DE3	Method	O104:H4	IAI39	K-12	Sakai	SE15	Santai	RM8426	
1,000	Expected	59.88	39.92	0.2*	0	0	0	0	
StrainFLAIR	56.48	43.52	0	0	0	0	0	
(0.995)	(0.989)	(0.254)	(0.189)	(0.151)	(0.192)	(0.214)	
Kraken2	38.93	60.76	0.11	0.05	0.08	0.04	0.03	
25,000	Expected	57.14	38.1	4.76*	0	0	0	0	
StrainFLAIR	54.12	41.72	4.16	0	0	0	0	
(0.995)	(0.989)	(0.584)	(0.266)	(0.177)	(0.282)	(0.298)	
Kraken2	37.75	58.93	2.16	0.28	0.34	0.25	0.29	
200,000	Expected	42.86	28.57	28.57*	0	0	0	0	
StrainFLAIR	46.96	35.32	17.72	0	0	0	0	
(0.993)	(0.988)	(0.711)	(0.318)	(0.211)	(0.346)	(0.351)	
Kraken2	31.14	48.83	13.53	1.57	1.67	1.58	1.68	

Interestingly, the proportion of detected specific genes for each strain (Fig. 4) seems to highlight a pattern allowing to distinguish—in this specific experiment - present strains, absent strains and likely new strains close to the reference in the graph. According to the experiments with enough coverage (from 25,000 simulated reads for BL21), three groups of proportions could be observed: proportion of almost 100% (O104:H4 and IAI39 : strains present in the mixtures and in the reference graph), proportion under 30–35% (Sakai, SE15, Santai, and RM8426 : strains absent from the mixtures), and an in-between proportion around 60–70% for K-12 (closest strain to BL21).

Figure 4 Proportion of detected specific genes for each simulated experiment with variable coverage of the BL21 strain, absent from the reference graph.

It was expected that an absent strain would have specific genes detected as StrainFLAIR detects a gene once only one read mappped on it. However, all absent strains had a proportion at around 30% except K-12 which proportion was twice higher. Conjointly with the non-null abundance estimated for the reference K-12, this suggests the presence of a new strain whose genome is highly similar to K-12.

Validation on a real dataset

We used a mock dataset available on EBI-ENA repository under accession number PRJEB42498, in order to validate our method on real sequencing data from samples composed of various species and strains. The mock dataset is composed of 91 strains of bacterial species for which complete genomes or sets of contigs are available, including plasmids. Among the species, two of them contained each two different strains. Three mixes had been generated from the mock, and we used the “Mix1A” in the following results.

Even though 20 out of 91 strains were absents in this mix, we indexed the full set of 91 genomes. This was done in order to mimic a controlled StrainFLAIR use case where the the reference graph contains a mix of strains present and absent in the queried data. The metagenomic sample was sequenced using Illumina HiSeq 3000 technology and resulted in 21,389,196 short paired-end reads.

We compared our results to the expected abundances of each strain in the sample defined as the theoretical experimental DNA concentration proportion. As such, it has to be noted that potential contamination and/or experimental bias could have occurred and affected the expected abundances.

Strain detection

Among the 91 strains used in the reference variation graph, StrainFLAIR detected 65 strains. All of these 65 strains were indeed sequenced in Mix1A. Hence, StrainFLAIR produced no false positive. From the 26 strains considered absent by StrainFLAIR, 20 were not present in the sample (true negatives) and 6 should have been detected (false negatives). However, the term false negative has to be soften as the ground truth remains uncertain. Among those 6 undetected strains, all of them had theoretical abundance below 0.1%.

More precisely, among the 6 strains undetected by StrainFLAIR, 5 had some detected genes, but below the 50% threshold. In this case, by default, StrainFLAIR discards these strains. Finally, only one of the undetected strains (Desulfovibrio desulfuricans ND 132) should have been theoretically detected (even if its expected coverage was below 0.1%), but no specific gene was identified. Considering that StrainFLAIR uses a permissive definition of detected gene (at least one read maps on the gene), having strictly no specific genes detected for Desulfovibrio desulfuricans ND 132 suggests that this strain might in fact be absent from Mix1A. This is also supported by the result from Kraken2 which estimated a relative abundance of ≈9E−5, almost 500 times lower than the theoretical result.

As in the simulated dataset validation, Kraken2 affected non-null abundances to all the references.

Strain relative abundances

For the estimated relative abundances, StrainFLAIR gave more similar results compared to the state-of-the-art tool Kraken2 than the experimental values (Fig. 5). The sum of squared error between StrainFLAIR and Kraken2 was around 11. StrainFLAIR and Kraken2 gave similar results compared to the experimental values, with sum of squared errors of around 209 and 211 respectively.

Figure 5 Experimental relative abundance compared to relative abundance as computed by StrainFLAIR and Kraken2.

A selection of relevant results is shown here, see Section S1.7 for the complete results. (A) Represents a case where StrainFLAIR and Kraken2 give similar results to the experimental value (18 cases over 91). (B) Represents a case where StrainFLAIR and Kraken2 give similar results, but lower than the experimental value (26 cases over 91). (C) Represents a case where StrainFLAIR and Kraken2 give similar results, but greater than the experimental value (16 cases over 91). (D, E, F, G) Represent the two species represented by two strains each. (H, I) Represent two atypical cases.

Interestingly, Thermotoga petrophila RKU-1 is the only case where results from StrainFLAIR and Kraken2 differs greatly, with, in addition, the theoretical abundance being in-between. Moreover, Thermotoga sp. RQ2 is the strain expected to be absent that Kraken2 estimates with the highest relative abundance among the other expected absent strains, and the only one exceeding the relative abundances of two present strains. Considering the previous results on the simulated mixtures and that Thermotoga petrophila RKU-1 and Thermotoga sp. RQ2 are close species (fastANI around 96.6%) it could be an additional indicator of how tools like Kraken2 can be mislead by too close species or strains.

In the sample, the species Methanococcus maripaludis was represented by two strains (S2 and C5) and the species Shewanella baltica likewise (OS223 and OS185). StrainFLAIR successfully distinguished and estimated the relative abundances of each strain of these two genomes. In this very situation and contrary to results on E. coli strains, Kraken2 was also able to correctly estimate the abundances.

Discussion

Recent advances in sequencing technologies have provided large reference genome resources. Representation and integration of those multiple genomes, often highly similar, are under active development and led to genome graphs based tools. Integrating multiple genomes from the same species is particularly interesting as it provides new opportunities to characterize strains, a key resolution. This taxonomic level can highlight new associations with diseases or with efficiency/toxicity of drugs for instance that the analysis at the species level currently masks. Particularly for gut microbiota studies, characterizing individual gut microbiota and targeting specific bacterial strains will open the field of precision medicine (Albanese & Donati, 2017; Marchesi et al., 2016).

In this context, we developed StrainFLAIR, a new computational approach for strain level profiling of metagenomic samples, using variation graphs for representing all reference genomes. Our intention was in the one hand to test whether or not indexing highly similar genomes in a graph enables to characterize queried samples at the strain level, and, in the other hand, to provide a end-user tool able to perform the indexing of genomes and the query of reads including the analyses of mapping results.

The method exploits state-of-the art-tools additionally to novel algorithmic and statistical solutions. By indexing microbial species and/or strains in a graph, it enables the identification and quantification of strains from a sequenced sample, mapped onto this graph.

Albeit in a controlled experiment simplifying the complex reality, we have demonstrated on simulated and on real datasets the ability of our method to identify and correctly estimate the abundance of microbial strains in metagenomic samples. In this context, StrainFLAIR was able to highlight the presence and also to estimate a relative abundance for a strain similar to existing references, but absent from these references.

We also showed that StrainFLAIR tended to set to zero the predicted abundance of low abundant strains, while a tool like Kraken2 was able detect them. As a result, it seemed that StrainFLAIR looses the ability to detect very low abundant strains. However, in our simulations, this situation corresponded to coverages of 0.03× or less, hence simulating a strain for which not all genomic content was present. Eventually, regarding this extremely low coverage, it might be more relevant to define this strain as absent. Overall, there is a need to distinguish between low abundant strains, insufficient sequencing depth, and reads from intergenic regions or other genes randomly matching genes. In this regard, StrainFLAIR integrated a threshold on the proportion of specific genes detected that can be further explored to refine which strain abundances are set to zero. Importantly, results also showed that our graph-based tool had no false positive call, contrary to general purpose tool Kraken2 that detected 100% of strains that were indexed but absent from queried reads.

From the validation on real datasets, we showed that StrainFLAIR was still able to correctly estimate the relative abundances in a more complex context mixing both different species and different strains, without being biased by references absent in the sample.

Our methodology taking into account all mapped reads and imposing a threshold that sets some strains abundances to zero seems more adequate and closer to what is expected (experimental data or ground truth) compared to other tools. Moreover, being able to detect some queried strains as absent is particularly interesting in the metagenomics context. Unlike mock datasets that are of controlled and known compositions, no prior knowledge is available for real metagenomic samples. They require the most exhaustive references—including unnecessary genomes—hence strains absent from the sample. StrainFLAIR is a new step towards the objective to take into account those unnecessary genomes without biasing the downstream analysis.

Measured computation time performances show that StrainFLAIR enables to analyse million reads in a few hours. Even if this opens the doors to routine analyses of small read sets, new development efforts will be made for reducing computation time in order to scale-up to very large datasets. Additionally, although StrainFLAIR showed convincing results on simulated and real datasets, exploring more complex situations is still necessary. First, the mock represented a controlled sample with prior knowledge for building the reference set. While this can be reproduced in a real situation by pre-filtering a genome database (using Kraken2 for example), further work might be needed to evaluate the scalability of our method with larger reference sets. However, we also showed that even by adding unnecessary genomes (absent from the queried sample) StrainFLAIR was able to correctly define them as absent strains. Secondly, we presented a case of one unknown strain in a mixture close to one of the reference strain. Future works will aim to address the issue of having several unknown strains close to the same reference or a mix of known and unknown strains close to the same reference, which StrainFLAIR can not distinguish yet.

Genomic plasticity and diversity is of increasing importance in microbiology, and lead to the field of pangenomics. Pangenomics can mainly be defined and explored in two ways. First, from the gene presence/absence perspective, also allowing to characterize core and accessory genome of a species. Secondly, from fine analysis of genomic variations. StrainFLAIR, which uses variation graphs to index clusters of genes, has the potential to cover both of those aspects. Indeed, graph structures, used as model for representing a set of related sequences, are then of great interest to capture all information on presence/absence of genes and variation/similarity of sequences, leading to new highlights on genome organization and regions of plasticity in a species. The variability provided by the sequencing of new genomes arises new challenges. In particular, this variability will need to be integrated into the graphs, which assumes a dynamic structure.

The natural continuation will be related to the dynamical update of the reference graph used with StrainFLAIR when novel species or strains are detected. As suggested in this work, when an indexed strain is detected in a query sample but with a low (≤75%) proportion of genes detected, this reflects the presence of another strain similar, but distinct. Other metrics could be used such as the mapping of non-colored paths of the graph and by nucleotidic variations between mapped reads and the graph sequences, and, of course, by non-mapped reads. Reads from these so-detected novel species or strains may be assembled using third-party haplotype-aware assemblers and the assembled sequences of genes will have to be added to the reference variation graph, updating clusters and path colors.

Supplemental Information

Supplemental Information 1 Supplementary Materials.

Click here for additional data file.

This work used the GenOuest bioinformatics core facility. We acknowledge Mircea Podar for the providing of the mock dataset in premium access. Finally, we thank Mahendra Mariadassou, Rayan Chikhi, Olivier Jaillon and David Vallenet for all their advice along this work.

Additional Information and Declarations

Competing Interests

Author Contributions

Data Availability

The authors declare that they have no competing interests.

Kévin Da Silva conceived and designed the experiments, performed the experiments, analyzed the data, prepared figures and/or tables, authored or reviewed drafts of the paper, and approved the final draft.

Nicolas Pons conceived and designed the experiments, authored or reviewed drafts of the paper, and approved the final draft.

Magali Berland conceived and designed the experiments, authored or reviewed drafts of the paper, and approved the final draft.

Florian Plaza Oñate conceived and designed the experiments, authored or reviewed drafts of the paper, and approved the final draft.

Mathieu Almeida conceived and designed the experiments, authored or reviewed drafts of the paper, and approved the final draft.

Pierre Peterlongo conceived and designed the experiments, authored or reviewed drafts of the paper, and approved the final draft.

The following information was supplied regarding data availability:

StrainFLAIR is available at http://github.com/kevsilva/StrainFLAIR.

The mock dataset is available at EBI-ENA: PRJEB42498.

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
