# Peer review of "StrainFLAIR: strain-level profiling of metagenomic samples using variation graphs"

_PeerJ, doi:10.7717/peerj.11884_

## Round 0.1 · original submission · Major Revisions

Dear Dr. Da Silva and colleagues:

Thanks for submitting your manuscript to PeerJ. I have now received two independent reviews of your work, and as you will see, the reviewers raised some concerns about the research. Despite this, these reviewers are optimistic about your work and the potential impact it will have on research studying bioinformatics approaches for strain-level processing of metagenomic samples. Thus, I encourage you to revise your manuscript, accordingly, taking into account all of the concerns raised by both reviewers.

The main concerns are a lack of sufficient background, missing datasets used in comparisons, and numerous edits to the text. Your work should be repeatable, so strive to provide everything necessary such that all assays can be repeated independently by others.

There are many minor suggestions to improve the manuscript.

Therefore, I am recommending that you revise your manuscript, accordingly, taking into account all of the issues raised by the reviewers.

Good luck with your revision,

-joe

Reviewer 1 ·

Basic reporting

I enjoyed reading the manuscript. It is well written and easy to follow. I have a concern about the presentation of background/literature survey. Strain level abundance estimation is well studied problem and there are numerous algorithms that are designed to achieve this. I didn't see these studies being mentioned in the background. For eg. tools such as StrainPhlan and DESMAN provide strain level abundance. I encourage authors to provide detailed literature review that covers recent advances in the field.

Experimental design

The methods are described in detail.

Although authors have presented the results on simulated and synthetic datasets, the results on the real metagenomic datasets are not provided. Understood, that the composition of such samples would be unknown, but uncovering the relative abundance of known strains with StrainFlair would be very helpful.

As alluded before, authors need to compare their method with some of the existing tools for strain abundance estimation to show that their method provides value over the existing methods. It can be in terms of higher accuracy, lower computer resource requirements or lower runtime. I understand that the authors have done the comparison with Kraken2, but the strain abundance estimation is not the primary use case for Kraken2.

Validity of the findings

The discussion section is well written.

Due to the lack of comprehensive benchmarking, the novelty or the benefit of the proposed method can not be assessed.

Reviewer 2 ·

Basic reporting

This manuscript presents StrainFLAIR, a promising approach to abundance estimation of bacterial strains in metagenomic samples. The article is well structured, with useful figures and interesting results. The introduction describes the relevance of the research, but a description of related work in the context of strain abundance estimation in metagenomics is missing. For example, I had expected methods like MetaPhiAn2, PanPhiAn, KrakenUniq and many more to be discussed. I was also surprised that the term “strain-level profiling”, which is part of the title of this manuscript, is not defined in the introduction (or anywhere else in the manuscript).

Some sections are well written, while others are rather vague or messy and require further clarification. To be more specific:
- The introduction starts with a discussion of linear reference genomes, stating that "the usage of reference 

genomes represented as flat sequences reaches some limits". I'm not sure which limits the authors are referring to.
- The methods section has several subsections, describing the different steps in the workflow. Figure 1 illustrates this nicely, but the overall workflow is not discussed in the main text. I think the methods section would benefit greatly from a description of figure 1, the algorithmic overview, at the beginning of the section. Because this is currently missing, it was unclear what "reads attribution", "path attribution" and "colored path attribution" mean (these are subsections later on in the methods). Also the content of these sections is quite vague.
- The description of simulated datasets is very lengthy and not very clear; I think it would be easier to follow if this section was more concise.
- The last two paragraphs of the discussion were also very vague and require further attention. I think the message can be very strong, but in its current form that is not clear.

I found the figures in this manuscript very helpful, in particular figure 3a. Also fig 3b was useful, but the caption requires some extra attention: the description is not very clear and I think the figure misses an illustration of the second step of the algorithm (last sentence of the caption).

The software and the simulated datasets are provided online, on a well-organized git repository. The only thing that I could not find were the exact commands to reproduce the results presented in the manuscript.

Experimental design

The methods are described in sufficient detail, but some parts of this section are too vague; please see my previous comment for further details and examples.

I have some concerns regarding the design of benchmarking datasets. The strains from which reads are simulated are fully identical to the references used, which is a highly unrealistic scenario. To truly evaluate performance of the proposed algorithm, one needs a dataset that is independent of the reference set used. If I understand correctly, the authors try to tackle this problem by excluding one of the strains from the reference set, but as a result, the algorithm predicts a different strain to be present. To me this seems like a wrong, misleading prediction. I would be very interested to see how the algorithm performs in a setting where the reference set was chosen independently from the simulated data, because this is closer to reality.

Validity of the findings

The results show that on simulated data the proposed algorithm outperforms Kraken2 in many aspects. On real data, however, this advantage disappears. I agree with the authors that it is difficult to evaluate performance on real data, as it does not come with a hard ground truth. But I'm not convinced by the simulated data sets for two reasons: first, because the references used are identical to the strains present, while it is reasonable to expect some mutations here; and second, because a threshold on the minimal number of genes to be present for a strain to have a prediction is set to 50%, but this seems to be based on results in Figure 4. So it looks like the method has been overfitted to this specific data set, making the results less convincing.

The authors compare their methods against one other method, Kraken2, which is among the state-of-the-art methods in metagenomic analysis. However, there may be better candidates as well: KrakenUniq, MetaPhiAn2, or DiTASiC, to name a few. I don't expect the authors to run every available method, but I'm not sure that Kraken2 is the best choice to compare against.

I appreciate the supplementary material, in particular the performance analysis (Table S1). I think this analysis should be discussed in the main paper, also including these statistics for other methods, as this is a crucial aspect of any bioinformatics workflow.

Additional comments

Some remaining comments/questions:

The manuscript uses both the terms indexation and indexing. I'm not familiar with indexation, but I guess the authors mean indexing? Since this is the common term in the field I would suggest to replace indexation by indexing throughout the manuscript.

L115: "minimap2 is used to generate a multiple sequence alignment". As far as I know, minimap2 does not perform multiple sequence alignment. I guess this sentence actually refers to all-versus-one alignment, from which a multiple sequence alignment is inferred?

L134: "continuous paths (successive nodes joined by only one edge)". This definition is not clear, it looks like you define either just paths in the graph, or simple paths.

L162: "the abundance is distributed to each matching colored path relatively to the ratio between them". Intuitively this does not make sense to me: for example, if the ambiguous alignments involve several colored paths, with some of them hardly any unique mappings because the paths are so similar, then the resulting coverages could be very misleading. Could you explain why this approach makes sense?

L179: "the gene abundance is estimated by the mean of the nodes abundance". Do you take node lengths into account by taking a weighted average here?

L180-182: "In order to not underestimate ... without the nodes of the path never covered by a read is also outputted". Could you also address this problem by looking at the median node abundance instead of excluding certain nodes?

L335: "a key resolution, for instance opening the field of precision medicine". Very vague statement, could you be more precise?

L363: "seems more adequate and closer to what is expected in reality" -> Compared to what?

---

## Round 0.2 · Minor Revisions

Dear Dr. Da Silva and colleagues:

Thanks for revising your manuscript. The reviewer is very satisfied with your revision (as am I). Great! However, there are some minor concerns still raised, and some edits to make. Please address these ASAP so we may move towards acceptance of your work.

-joe

Reviewer 2 ·

Basic reporting

The authors have addressed many of my concerns satisfactorily: the manuscript reads very well, the background and aims are clear and the methods are well described. Supplementary results now include benchmarking results on other tools, which is very informative. However, I still have some major concerns regarding the experimental design and conclusions, which I will explain below.

Experimental design

In metagenomic profiling, the sample composition is unknown. Hence, the reference set cannot be the same as the queried genomes, it has to be much larger. In line 319, regarding the experiments on real data, the authors write that all known 91 strains were included "to mimic a classical StrainFLAIR use case where the queried data is mainly unknown". But this still does not reflect a realistic scenario, because one would not know the selection of 91 strains beforehand. In general, the databases used for such analysis are huge. Would StrainFLAIR be able to index and make predictions for thousands of genomes, how does it scale? And what do the predictions look like if one uses a larger reference set?

L341: "Kraken2 affected non-null abundances to all the references and thus could not be used to definitely conclude on presence/absence of strains in this sample". Also a similar statement in line 389-390. However, commonly, one applies a minimal abundance threshold to filter out such false positive calls, so these statements are not entirely fair. Moreover, I could not find a description of how the relative abundances are computed from the read counts. Are read counts normalized by genome size or aligned length?

This also relates to the colored path attribution step in StrainFLAIR: shouldn't unique read counts be normalized by alignment length here, before computing the ratio between colored paths? Or is it the ratio of alignment paths? It would be helpful to distinguish more clearly between alignment paths and reference paths, instead of just "paths" which can mean either.

Validity of the findings

Although the conclusions and discussion have improved considerably, there is one claim (made at several points in the manuscript) with which I strongly disagree:
L87-88: "Notably, we could also identify close strains not present in the reference"
L293: "Labelled as K-12, it also gave close estimations for BL21"
L300-301: "the proportion of detected specific genes for each strain seems to highlight a pattern allowing to distinguish present strains absent strains and likely new strains close to the reference"
L377-379: "StrainFLAIR was able to highlight the presence and also to estimate a relative abundance for a strain similar to existing references, but absent from these references"

I see that in these specific data sets, the proportion of detected genes provides an indication of the similarity of the true strain to the predicted strain. But what if the (falsely) predicted strain (in this setting K-12) is also present? And what if there are multiple unknown strains? Or if the unknown strain is equally similar to multiple references? If I understand the approach correctly, you would not be able to tell the difference. In reality there will be many genomes that are not present in your reference set, which would affect the predicted abundances substantially, and I don't see how the proportion of detected genes will help with this. I find a statement like "StrainFLAIR can identify strains not present in the reference" misleading and it seems that using an incomplete reference set can lead to highly skewed abundances due to false positives, like the K-12 predictions in the absence of any K-12 in the sample.

What I also don't understand is why the authors do not include DiTaSic results (from the supplement) in the manuscript. Table S5 shows that these results are far better than Kraken2 and also better than StrainFLAIR. What is the reasoning for not including these results?

Additional comments

Some minor textual things:

L30: "later" -> latter

L72: "... are based on de novo approaches using known core genes and a single reference genome" - I'm confused as to why you call this de novo of it's using a reference genome?

L98: I think it would be helpful to specify here that you mean strain-specific reference genomes

L163: "In the meanwhile, ..." -> Meanwhile, ...

In many places in the manuscript you write about "specific genes". But what do you mean by "specific"?

---

## Round 0.3 · accepted · Accept

Dear Dr. Da Silva and colleagues:

Thanks for revising your manuscript based on the concerns raised by the reviewer. I now believe that your manuscript is suitable for publication. Congratulations! I look forward to seeing this work in print, and I anticipate it being an important resource for groups studying bioinformatics approaches for strain-level processing of metagenomic samples. Thanks again for choosing PeerJ to publish such important work.

Best,

-joe

Reviewer 2 ·

Basic reporting

No comment.

Experimental design

No comment.

Validity of the findings

No comment.

Additional comments

Thank you for addressing my questions and concerns, I have no further comments.